# Efficient Adsorption of Ammonia by Surface-Modified Activated Carbon Fiber Mesh

**DOI:** 10.3390/nano13212857

**Published:** 2023-10-28

**Authors:** Yongxiang Niu, Chao Zheng, Yucong Xie, Kai Kang, Hua Song, Shupei Bai, Hao Han, Shunyi Li

**Affiliations:** 1School of Ecology and Environment, Zhengzhou University, Zhengzhou 450001, China; xznyxmx@163.com; 2State Key Laboratory of NBC Protection for Civilian, Beijing 102205, China; 13263157913@163.com (C.Z.); ycong6364@163.com (Y.X.); kangkai@sklnbcpc.cn (K.K.); huasww@163.com (H.S.); baisp@263.net (S.B.)

**Keywords:** ammonia capture, activated carbon fiber, surface modification, chemisorption, equivalent heat of adsorption

## Abstract

In view of the characteristics and risks of ammonia, its removal is important for industrial production and environmental safety. In this study, viscose-based activated carbon fiber (ACF) was used as a substrate and chemically modified by nitric acid impregnation to enhance the adsorption capacity of the adsorbent for ammonia. A series of modified ACF-based adsorbents were prepared and characterized using BET, FTIR, XPS, and Boehm titration. Isotherm tests (293.15 K, 303.15 K, 313.15 K) and dynamic adsorption experiments were performed. The characterization results showed that impregnation with low concentrations of nitric acid not only increased the surface acidic functional group content but also increased the specific surface area, while impregnation with high concentrations of nitric acid could be able to decrease the specific surface area. ACF-N-6 significantly increased the surface functional group content without destroying the physical structure of the activated carbon fibers. The experimental results showed that the highest adsorption of ammonia by ACFs was 14.08 mmol-L^−1^ (ACF-N-6) at 293 K, and the adsorption capacity was increased by 165% compared with that of ACF-raw; by fitting the adsorption isotherm and calculating the equivalent heat of adsorption and thermodynamic parameters using the Langmuir–Freundlich model, the adsorption process could be found to exist simultaneously. Regarding physical adsorption and chemical adsorption, the results of the correlation analysis showed that the ammonia adsorption performance was strongly correlated with the carboxyl group content and positively correlated with the relative humidity (RH) of the inlet gas. This study contributes to the development of an efficient ammonia adsorption system with important applications in industrial production and environmental safety.

## 1. Introduction

Ammonia is a colorless gas with a pungent odor mainly produced by agricultural activities, industrial processes, and transportation [1]. As a typical pollutant with strong corrosive alkaline properties, it poses significant hazards to human health [2] and the atmospheric environment. The Acute Exposure Guideline Levels (AEGL) set by the USA Environmental Protection Agency specify a safe concentration of 30 ppm for ammonia exposure during a time frame of 10 min to 8 h [3]. Ammonia not only contributes to the formation of haze in the atmosphere [4] but also causes soil acidification and eutrophication of water bodies [5]. Furthermore, ammonia gas can lead to equipment corrosion in processes associated with its use [6,7], thereby posing substantial safety risks. In cases of ammonia leaks or accidents, as well as in environments with high ammonia concentrations exceeding safety standards, it is important to implement confined space protection measures. In all of the above cases, there is significant interest in and demand for the development of air filtration units that can effectively adsorb NH_3_.

Adsorption, with its high efficiency and low energy consumption, is widely used in ammonia separation processes [8]. Adsorbents used for ammonia purification and recovery include zeolites [9], activated carbon [10], molecular sieves [11], metal–organic framework (MOF) materials [11], and porous organic polymers [12]. Activated carbon fiber (ACF) is a porous fibrous adsorbent carbon material prepared by high-temperature carbonization and activation of organic fibers with controllable porosity, high activity, and abundant surface chemical properties, and it has been widely studied for the adsorption of NH_3_ at room temperature [13]. Compared to traditional activated carbon, ACF has significant advantages, such as a large number of micropores [14,15], high temperature resistance, and ease of processing into various shapes [16]. These properties enable ACF to have faster adsorption rates, higher adsorption capacities [17], and broader application areas. Furthermore, a study has demonstrated that activated carbon fibers exhibit electrical conductivity [18]. This property allows for the generation of heat through energization, enabling efficient regeneration of adsorbed gases [19]. In the design of purification units, apart from adsorption capacity, tightness and pressure drop are crucial factors to consider. While GACs exhibit slow kinetics due to their large particle size, PACs and activated carbon fiber cloths offer fast mass transfer. However, the significant drawback of PACs and activated carbon fiber cloths is their extremely high pressure drop. Conversely, the mesh structure of ACFM provides advantages, such as fast kinetics, low pressure drop, and simplified accommodation, making it a favorable choice for various applications [20]. In addition, the mesh structure of ACFM offers lower resistance and requires less energy compared to ACFC.

Both the pore structure and surface chemistry of ACF have an effect on ammonia adsorption. The micropore size of ACF is slightly larger than the average molecular dynamics diameter of NH_3_ molecules (0.365 nm), which is favorable for capturing ammonia. Due to the presence of a lone pair of electrons in NH3, the adsorption process can be facilitated by the Lewis and Brønsted acid sites provided by acidic functional groups on the surface (carboxyl, phenolic hydroxyl, ester groups, etc.) [21]. However, due to the limitations of carbon-based materials [22], ACF lacks sufficient surface acidity, and, therefore, only weakly adsorbs NH_3_ through H-bonding or Lewis acid–base interactions [23]. By modifying and designing ACF with functional groups, new materials with better NH_3_ adsorption performance and higher stability can be derived. Because NH_3_ has a very high vapor pressure (10 atm at 298 K), the physical adsorption of activated carbon is thermodynamically unfavorable [24]. Therefore, current modification methods mainly focus on improving chemical adsorption by optimizing surface functional groups of activated carbon fibers to enhance the interaction between the surface and NH_3_ molecules. The main methods include oxidation by chemicals to generate acidic oxygen-containing functional groups capable of capturing alkaline NH_3_, as well as loading metal oxides and chlorides to promote ammonia adsorption through complexation reactions [22]. Chen [18] et al. oxidatively modified ACFC by nitric acid to increase the surface acidic oxygenated functional group content and electrically regenerate it. Qajar et al. [25] reported that the NH_3_ adsorption capacity of synthetic non-porous carbon material increased from 180 mg·g^−1^ to 306 mg·g^−1^ after treatment with nitric acid at 298 K. In 2016, Zheng et al. [26] prepared carbon fiber composite materials from phenolic precursors, and the NH_3_ adsorption capacity increased from 10 mg·g^−1^ to 50 mg·g^−1^ after oxidation with concentrated HNO_3_. Xu et al. [15] prepared a series of activated carbon fiber membrane adsorbents simultaneously loaded with metal chlorides and metal oxides, achieving an NH_3_ adsorption capacity of 22.5 mg·g^−1^.

This study aimed to enhance the adsorption efficiency of ammonia (NH_3_) on activated carbon fiber mesh (ACFM) utilized in electrothermal regeneration studies. Additionally, this study aimed to examine the impact of surface chemistry on the adsorption process. In this study, ACF was treated with different concentrations of nitric acid. The changes in the pore structure of ACF after nitric acid modification were investigated using BET analysis. The surface functional groups of ACF were characterized using FTIR, XPS, and Boehm titration to analyze the effect of nitric acid modification on the content of surface functional groups. The adsorption–desorption isotherms of ammonia on both the original ACF and modified ACF were measured at different temperatures (293.15 K, 303.15 K, 313.15 K). The Langmuir–Freundlich adsorption equilibrium model was used to describe the adsorption isotherms and calculate the isosteric adsorption heat of the materials. In order to analyze the dynamic adsorption process of ACF, dynamic adsorption experiments of ammonia on ACF were conducted in a dynamic tube. The breakthrough curve of ammonia on ACF was measured to evaluate the adsorption performance of ACF. The breakthrough curves of ACF adsorbing ammonia under different moisture content conditions were also tested to analyze the influence of water vapor on ammonia adsorption. Finally, the correlation between the number of acidic oxygen-containing functional groups and adsorption performance was analyzed to explore the mechanism of improved ammonia adsorption performance on modified ACF. Based on the current lack of work on the modification of activated carbon fiber mesh for electrothermal regeneration, this study improved the adsorption capacity of viscose-based ACFM for ammonia and further optimized the capacity of the ACF adsorption–electrothermal regeneration unit for ammonia, which contributes to the achievement of efficient and durable protection against ammonia.

## 2. Materials and Methods

### 2.1. Materials

All of the viscose-based ACF used in this experiment was purchased from Zhengzhou Zhongsida Environmental Science and Technology Co., Ltd. (Zhengzhou, China), woven from viscose fiber raw silk into blanks, and then processed from the blanks into an activated carbon fiber mesh (ACFM), including the weaving process, flame-retardant impregnation, low-temperature charring, high-temperature charring, and activation, resulting in the formation of the mesh structure, with a surface area of about 1000 m^2^·g^−1^.

### 2.2. Preparation of the Adsorbents

To increase the amount of oxygen-containing functional groups, the prepared ACFMs were treated with nitric acid (HNO_3_) in a procedure similar to that of Shan [27] et al. One hundred 5 × 5 cm^2^ pieces of ACFM were cut, and the weight of each piece was controlled to be 0.278 ± 0.01 g. Each ACFM sample was treated with 500 mL of nitric acid at different concentrations for 12 h at room temperature. Then, the samples were washed several times with 500 mL of ultrapure water until the pH of the wash water reached 4. After that, the samples were dried in an oven at 110 °C for more than 12 h to finally obtain the modified ACFM. The unmodified ACF is labelled ACF-raw, and the material obtained by acid impregnation is labelled ACF-X-A, where X shows the acid used for impregnation (N stands for nitric acid, H stands for hydrochloric acid, and P stands for phosphoric acid) and A shows the concentration of nitric acid used for impregnation (mol-L^−1^); as such, the sample obtained using 6 mol-L^−1^ nitric acid in the modification is labelled ACF-N-6.

### 2.3. Characterization

Nitric-acid-impregnated samples were characterized by testing CO_2_ adsorption and desorption isotherms at 273 K using a Quantachrome Nova 4000e (Quantachrome, Boynton Beach, FL, USA) device. The BET specific surface area was calculated by fitting the BET equation to the CO_2_ isotherm. Density Functional Theory (DFT) was used to calculate the micropore volume (pore size < 2 nm), total pore volume, and pore size distribution of the micropores. The relative oxygen content and oxygen-containing functional groups on the sample surface were determined through X-ray photoelectron spectroscopy (Thermo Scientific K-Alpha, Thermo Fisher Scientific, Waltham, MA, USA). The position of the Cls peak was calibrated to 284.8 eV. The relative content of each oxygen-containing functional group was expressed as a percentage of its peak area. The content of surface acidic oxygenated functional groups (i.e., phenols, lactones, and carboxylic acids) on the samples were determined using Boehm titration. The functional groups of the raw and modified materials were determined through FTIR (Thermo Scientific Nicolet iS20, Thermo Fisher Scientific, MA, USA). The samples were thoroughly ground after mixing with KBr (mass ratio 1:150) and pressed to form translucent sheets prior to analysis.

### 2.4. Measurement of Ammonia Adsorption Isotherms

The adsorption/desorption isotherms of ammonia (NH_3_) on ACF were tested under three room temperature conditions (T = 293.15 K, 303.15 K, 313.15 K) using a Corrosive Gas Adsorption Analyzer (BSD-PMC, Beijing Best Instruments (Beijing, China)). Adsorption/desorption isotherms were tested on three molecular sieves. The test pressure range was 0–1.0 bar, and the sample pretreatment conditions were 383.15 K and vacuum degassing treatment for 12 h. The test pressure range was 0–1.0 bar.

### 2.5. Adsorption Equilibrium and Heat of Adsorption

The Langmuir adsorption isotherm is a simple model based on the assumption that adsorption occurs in a monolayer. The Freundlich adsorption isotherm is an empirical model describing non-ideal and reversible adsorption. The single-point Langmuir–Freundlich equation (Equation (2)) combines a form of the Langmuir and Freundlich equations and shows good utility for fitting data [28]. Equation (2) is often used to fit adsorption data on activated carbon and MOFs. It also reflects the complex adsorption interactions between the adsorbent and the adsorbate.
(1)$$\;n=n_{0}\times \frac{{\left(b\times p\right)}^{q}}{1+{\left(b\times p\right)}^{q}}\;$$
where *n* is the saturated adsorption capacity (mL·g^−1^); *n*_0_ is the maximum saturated adsorption capacity (mL·g^−1^); *b* is the factor less affinity constant; *q* is the exponent; and *p* is the relative partial pressure.

Through fitting the Freundlich adsorption isotherm equation, the adsorption model constants $k_{f}$ and $n_{f}$ are obtained. The Gibbs free energy (${\Delta G}^{{}^{\circ}}$), enthalpy change (${\Delta H}^{{}^{\circ}}$), and entropy change (${\Delta S}^{{}^{\circ}}$) of the adsorption system during the adsorption process of ammonia gas are studied. Understanding the thermodynamic properties of adsorption is crucial for explaining the adsorption mechanism, designing adsorbents, and optimizing adsorption processes.
(2)$$q_{e}={\;k}_{f}C_{e}^{n_{f}}$$
(3)$$k_{e}^{{}^{\circ}}=k_{f}^{1/n_{f}}$$
(4)$${\Delta G}^{{}^{\circ}}=-\mathrm{RTln}k^{{}^{\circ}}$$
(5)$${\Delta G}^{{}^{\circ}}={\Delta H}^{{}^{\circ}}-T{\Delta S}^{{}^{\circ}}$$
(6)$$\ln K^{0}=-\frac{{\Delta H}^{{}^{\circ}}}{R}\times \frac{1}{T}+\frac{{\Delta S}^{{}^{\circ}}}{\;R}$$
where Equation (6) is the Van’t Hoff equation. $k_{f}$ and $n_{f}$ are the Freundlich adsorption constants. R is the ideal gas constant (8.314 mol^−1^·k^−1^). *T* is the absolute temperature (K). ${\Delta G}^{{}^{\circ}}$ is the Gibbs free energy (KJ·mol^−1^). ${\Delta H}^{{}^{\circ}}$ represents the enthalpy change (KJ·mol^−1^), and ${\Delta S}^{{}^{\circ}}$ represents the entropy change(J·mol^−1^·K^−1^).

The Clausius–Clabellon equation (abbreviated: C-C equation) [29] was used to calculate the equivalent heat of adsorption of ammonia (NH_3_) on ACF-raw and HNO_3_-treated ACF.
(7)$$Q_{iso}=\frac{RT_{1}T_{2}}{T_{2}-T_{1}}ln\frac{P_{2}}{P_{1}}\;$$

$Q_{iso}$ is the heat of adsorption (J·mol^−1^); $T_{1}$ and $T_{2}$ are the system temperatures (K); $P_{1}$ and $P_{2}$ are the absolute pressures (bar) corresponding to $T_{1}$ and $T_{2}$, respectively; and R is the ideal gas constant (8.314 mol^−1^·k^−1^).

### 2.6. Adsorption Dynamics

Dynamic testing was conducted in both a power tube and a fixed bed to evaluate the ammonia adsorption capacity. When testing the effect of moisture content on the adsorption process, a self-made fixed bed was filled with 100 layers of 5 × 5 cm^2^ activated carbon fiber (ACF) with a height of 5 cm, and the airflow rate was controlled at 10 L·min^−1^ using a flowmeter. When testing the ammonia adsorption performance of ACF modified with different concentrations of acid, 10 layers of ACF with a height of approximately 1.0 cm were placed in a cylindrical quartz tube with an inner diameter of 2 cm, and the total airflow rate was controlled at 950 ± 10 cm^3^-min^−1^ using a mass flow controller. A mixing chamber was added before entering the dynamic tube or fixed bed to improve the uniformity of the ammonia concentration, and the inlet concentration was maintained at 300 ppm. An infrared detector (QGS-08C, Beijing BAIF-Maihak Analytical Instrument Co., Ltd., Beijing, China) was used as the ammonia detector to monitor the ammonia concentration in the exhaust gas. When the outlet ammonia concentration reached 5% of the ammonia concentration in the inlet gas, it was considered breakthrough. When the outlet concentration reached 100% of the ammonia concentration in the inlet gas and remained stable without any changes for 10 min, it was considered saturated. The specific process is shown in Figure 1.

## 3. Results and Discussion

### 3.1. Physical Properties

The surface area and pore characteristics of materials are often determined using N_2_ adsorption/desorption isotherms at 77 K using the Brunauer–Emmett–Teller (BET) theory and density functional theory methods (DFT). However, there are kinetic limitations at low temperatures (77 K), where N_2_ molecules have difficulty diffusing into ultra-micropores and smaller micropores within a short period of time, leading to deviations in the analysis results. On the other hand, CO_2_ molecules have a smaller kinetic diameter and a very high saturation vapor pressure at 273 K. In this condition, gas diffusion is faster, and CO_2_ molecules can enter micropores below 0.4 nm. Therefore, CO_2_ has become an effective probing molecule for studying extremely narrow micropores in carbon materials.

The CO_2_ adsorption–desorption isotherms (a) and pore size distribution (b) of ACF-raw and HNO_3_-treated ACFs are shown in Figure 2. At 273 K, the curve depicting the changes in the amount of adsorbed CO_2_ with respect to pressure exhibits the characteristics of an I-type isotherm, as per the classification established by IUPAC. An I-type isotherm is typically indicative of an adsorbent material with a microporous structure. Figure 2b illustrates that the pore structure of all samples predominantly consisted of micropores. The primary pore size distribution of ACF ranged from 0.3 to 1.0 nm, with a pore size of 0.548 nm accounting for the highest percentage of pores. The specific surface area of micropores in ACF varied from 992.6 to 1056.4 m^2^·g^−1^. After HNO_3_ impregnation, the specific surface area and micropore volume of ACF slightly increase. With the increase in nitric acid concentration, the micropore volume gradually increases, and there is no significant change in the average pore size. However, when the concentration reaches 8 mol·L^−1^, the micropore volume starts to decrease. This could be due to the strong oxidizing nature of HNO_3_, which may cause damage to the pore walls of ACF, thereby reducing the specific surface area and pore volume and affecting the adsorption performance. Therefore, it is important to choose an appropriate concentration of the oxidizing agent during the modification process.

### 3.2. Surface Chemistry

The surface chemistry of ACF plays a key role in the adsorption of NH_3_ in activated carbon [22,23]. By oxidizing carbon materials using nitric acid, the content of oxygen functional groups, including carboxyl, phenolic, and lactone groups, can be increased [30]. These functional groups have the potential to significantly improve the adsorption capacity of NH_3_ [31,32].

Figure 3 shows the FTIR spectra of ACFs. In oxidized carbons, broad bands in the range of 1000–1300 cm^−1^ (with maxima at 1100–1200 cm^−1^) are commonly observed. These bands are attributed to the stretching vibrations of C-O bonds, which can be present in acids, alcohols, phenols, ethers, and esters groups [33]. The peaks observed at approximately 1600 cm^−1^ can be attributed to the presence of C=C bonds in the carbon matrix of the original ACF [26]. After nitric acid oxidation, HNO_3_-treated ACFs showed an increase in peak intensity near 1700 cm^−1^, which indicated the introduction of a variety of functional groups, including carboxylic acids, lactones [34], carbonyls, quinones [35], etc., and the wavenumber exceeding 1750 cm^−1^ may indicate anhydrides, especially under 8 mol·L^−1^ nitric acid treatment [10]; in general, the acid–base interactions of Brønsted are reversible, whereas amide or imide bonding is irreversible in the absence of a catalyst [30], which is not conducive to subsequent regeneration of the material but can be applied to the adsorption of ammonia at elevated temperatures. The peak of 1400 cm^−1^ belongs to the C-O stretching vibration on COO- [36]. The broad absorption band observed at 3300–2500 cm^−1^ is typically assigned to the stretching vibrations of O-H bonds, which are commonly derived from carboxylic functional groups [37]. In addition, the more pronounced peaks at 3400 cm^−1^ and 1040 cm^−1^ indicate that the number of carboxylic acid groups bound in the matrix is greatly increased [26].

The surface functional groups of ACF-raw and HNO_3_-treated ACF were further analyzed using XPS, and the XPS spectra survey scans of ACF-raw and HNO_3_-treated ACFs are shown in Figure 4. Two main peaks are observed in the scanning spectrum, which are attributed to C1s and O1s, respectively. The key design parameter for high-capacity adsorbents is the O/C ratio [38], which is associated indirectly with the number of surface groups present on ACF. High oxygen content is also typically considered to be the main reason for the high ammonia adsorption capacity [35], and it can be seen from Table 1 that the O/C ratio gradually increases from 14.72% to 19.19% with the increase in concentration of nitric acid used for impregnation, which indicates that the number of oxygen-containing functional groups in the pore structure gradually increases.

C1s spectra can be generally fitted by five characteristic peaks: graphite peak (284.8 eV) [15], C-O in phenols or ethers (286.2 eV), C=O in carbonyls (287.5 eV), and O-C=O in carboxyl and its derivatives (288.8 eV) [33]. The relative content of each functional group was calculated by dividing the corresponding area of small peaks by the total area, as shown in Table 2. The relative content of carboxylic acid and its derivatives and the relative content of phenolic or ether compounds increased from 12.49% to 16.13% and from 9.90% to 16.13%, respectively, among the acidic oxygen-containing functional groups on the surface of the modified samples. It can be seen from Figure 4 that both carboxylic acid and its derivatives and phenolic hydroxyl increased with the increase in the impregnation of the HNO_3_ concentration, whereas the relative content of the graphitic carbon showed a decrease, indicating that the increase in the concentration of impregnated nitric acid favors the formation of oxygen-containing functional groups, such as carboxyl and phenolic hydroxyl groups, of the sample.

In order to quantify the actual content of oxygenated acidic functional groups, Boehm titration was further used to determine the actual content of oxygenated functional groups. The titration results (Figure 5) of untreated ACFs and HNO_3_-treated ACFs are given in Figure 5, from which it can be found that the acidic functional groups, such as hydroxyl, lactone, and carboxyl groups, on the surface of ACF increased after HNO_3_ treatment, which corresponds to the FTIR and XPS results. It should be noted that the Boehm titration method does not differentiate between anhydrides and carboxylic acids. Consequently, the reported value of 1.48 mmol/g includes both functional groups, as indicated by the presence of anhydrides in the FTIR analysis [26].

### 3.3. Adsorption Equilibrium and Heat of Adsorption

#### 3.3.1. Ammonia Adsorption Isotherm Experiments

The adsorption/desorption isotherms of ammonia (NH_3_) on ACF-raw and HNO_3_-treated ACFs at different temperatures (T = 293.15 K, 303.15 K, 313.15 K) are shown in Figure 6 (absolute pressure as the horizontal coordinate), which were basically of the “Langmuir” type (classified as type I by IUPAC). The I-type isotherm is typically indicative of a microporous adsorbent. As shown in Figure 6, a rapid increase in ammonia adsorption occurs in the pressure range of P < 0.05 bar, while the growth rate of ammonia adsorption decreases in the pressure range of 0.05 bar < P < 1 bar. As shown in Figure 6a–e, the adsorption capacities of these adsorbents for ammonia at 293.15 K and 303.15 K were lower than their adsorption capacities at 278 K, especially under low partial pressure conditions. This is because the adsorption of ammonia on ACF is an exothermic process, and as the temperature increases, the molecular diffusion rate and surface adsorption energy increase. From the graph, it can be observed that the adsorption curve and desorption curve do not overlap and are not closed, indicating that ammonia forms strong chemical bonds with the adsorption sites, resulting in a slow desorption rate and incomplete desorption [39]. It can be assumed that chemisorption processes are present in the adsorption of ammonia on ACF. In addition, the adsorption performance of ammonia does not strictly correlate with the order of specific surface area and micropore size of the materials listed in Table 2. This indicates that the adsorption of ammonia on both raw ACF and nitric-acid-treated ACF is not solely determined by physical adsorption (van der Waals forces) but also involves chemical adsorption with certain surface functional groups. The adsorption capacity is regulated by both physical and chemical adsorption. Based on the characterization results, the increase in the number of surface acidic oxygen-containing functional groups after nitric acid modification enhances the adsorption sites with Brønsted and Lewis acid centers that can react with ammonia gas [30]. In this case, chemical adsorption is enhanced. Therefore, compared to raw ACF, nitric-acid-treated ACF exhibits significantly increased adsorption of ammonia.

Adsorption data of different materials at 293.15 K are shown in Figure 6f. The ammonia adsorption amounts of ACF-raw and HNO_3_-treated ACF were 273 mL·g^−1^, 305 mL·g^−1^, 301 mL·g^−1^, 311 mL·g^−1^, and 301 mL·g^−1^, respectively. Comparison revealed that ACF-N-6 had the maximum adsorption capacity. NH_3_ capture performances on various porous materials are summarized in Table 3. We can see that the ammonia adsorption capacity of ACF-N-6 is lower than that of some high-performance MOFs, but it is significantly higher than conventional materials, such as zeolite and bentonite. It also exhibits excellent performance among carbon materials. However, the application of MOFs is limited by their higher production costs compared to other commercial products, despite their exceptional performance in NH_3_ removal.

#### 3.3.2. Modelling of Ammonia Adsorption Equilibrium Isotherms

In order to illustrate the adsorption equilibrium on ACF-raw and HNO_3_-treated ACF of ammonia, this adsorption process is simulated using the Langmuir–Freundlich adsorption isotherm model.

Adsorption equilibrium modeling is the classical method for describing adsorption isotherms, and the higher the correlation between the equation and the data, the lower the error [45]. We used Langmuir–Freundlich modelling (Equation (1)) to fit the adsorption isotherms of ammonia (NH_3_) on ACF-raw and HNO_3_-treated ACF under the temperature conditions of 293.15 K, 303.15 K, and 313.15 K (absolute pressure as the abscissa), and the residuals were calculated (Residuals = experimental adsorption values—same fitted value at pressure). The result is shown in Appendix A. For the above materials, the correlation coefficients (R^2^) of the equation fits were all greater than 0.95 (Appendix A), and the errors were mainly concentrated in the lower pressure range (0 < P < 0.05 bar), which may be related to the binding strength and binding form of the surface adsorption sites and ammonia. The results showed that the Langmuir–Freundlich equation demonstrated a good fit to both ammonia adsorption isotherms and could be used for the prediction of adsorption data.

#### 3.3.3. Equivalent Heat of Adsorption and Thermodynamic Parameters

The heat of adsorption is an effective expression of the adsorption strength of ACF on ammonia [13]. When calculating the equivalent heat of adsorption, it is necessary to choose the equation with more accurate fitting results. According to the fitting results obtained from the Langmuir–Freundlich equation, the equivalent heat of adsorption was calculated using Equation (7). Figure 7 shows the calculated equivalent heat of adsorption curves versus the amount of ammonia adsorbed in the range of 0–250 mL·g^−1^. The heat of adsorption of ammonia on HNO_3_-treated ACF in the initial stage was higher than that of ACF-raw, which indicated that the initial adsorption stage of ammonia on HNO_3_-treated ACF was stronger than that of ACF-raw, which was consistent with the results of the acidic functional group content on the surface and represented a stronger chemisorption [46]. The thermodynamic parameters for ACF-raw and ACF-N-6 were calculated using the Equations (2)–(6). As shown in Table 4, both ΔG < 0, indicating that the adsorption behavior of ammonia gas on ACF is spontaneous. Both ΔH < 0, indicating that the adsorption process is exothermic. Both ΔS values are negative, indicating a decrease in system disorder after the adsorption process.

As shown in Figure 7, the heat of adsorption of ACF-N-6 was higher than that of the other adsorbents, indicating that the adsorption strength of ammonia on ACF-N-6 was greater than that of the other adsorbents. This is consistent with the aforementioned increase in the content of surface acidic oxygen-containing functional groups, suggesting that the increase in acidic oxygen-containing functional groups promotes the chemical adsorption of ammonia on the ACF surface. As the adsorption proceeded, the heat of adsorption of ammonia on ACF-N-6 gradually decreased to 40 kJ·mol^−1^, indicating that the chemical adsorption dominated the entire adsorption process [47]. The curves of ACF-N-2 and ACF-N-4 showed that before the heat of adsorption reached 30 mL·g^−1^ and 55 mL·g^−1^, respectively, the heat of adsorption was greater than 40 kJ·mol^−1^, indicating that chemical adsorption dominated. Afterward, ammonia condensed in the pores by capillary condensation, and the heat of adsorption decreased below 40 kJ·mol^−1^, indicating that physical adsorption dominated, mainly through van der Waals forces, similar to the entire process of ACF-raw. By comparing the four curves and combining them with the aforementioned study of changes in functional group content, it can be concluded that the increase in surface acidic oxygen-containing functional group concentration increases the ammonia adsorption sites on the ACF surface, allowing more acidic oxygen-containing functional groups to combine with ammonia through the Lewis and Brønsted acid–base reaction.

### 3.4. Ammonia Adsorption

#### 3.4.1. Breakthrough Tests

Dynamic breakthrough tests were employed to evaluate the efficiency of the ACF for removing NH_3_ from the air in a flow-through configuration. Hydrochloric-acid-modified samples and phosphoric-acid-modified samples were selected for comparison with nitric-acid-modified samples. The results are summarized in Figure 8.

Based on the results shown in Figure 8, it is evident that acid-modified ACF with varying concentration gradients can effectively adsorb ammonia gas from the airflow. Under similar conditions, NH_3_ started to break through at 7 min on ACF-raw. As the concentration of nitric acid (HNO_3_) and hydrochloric acid (HCl) in the modification reagents increases, the breakthrough time gradually lengthens. Among these, ACF-N-6 and ACF-H-6 exhibit the most optimal adsorption performance, with breakthrough times of 23 min and 13 min, respectively. Conversely, as the concentration of phosphoric acid (H_3_PO_4_) increases, the breakthrough time decreases. ACF-P-2 demonstrates the best adsorption performance, with a breakthrough time of 12 min. In conclusion, the modified ACFs exhibit improved adsorption performance for ammonia gas, with ACF modified with nitric acid showing the most effective adsorption effect.

#### 3.4.2. Ammonia Adsorption in Gas Streams with Different Moisture Contents

Water vapor significantly affects the adsorption of ammonia [48]. The breakthrough curves of ACF-raw and ACF-N-6 were tested under different inlet humidities (RH = 17%, 42%, and 68%), as shown in Figure 9. It can be observed that the breakthrough time and adsorption amount of ammonia for both ACF-raw and ACF-N-6 increased with increasing inlet humidity. At RH = 68%, the breakthrough adsorption capacity for ACF-raw and ACF-N-6 increased by 2.57 mg·g^−1^ and 3.76 mg·g^−1^, respectively, compared to dry conditions. Furthermore, the adsorption capacity under both wet and dry conditions was positively correlated with the moisture content of the air stream, with a steeper increasing trend observed for ACF-N-6.

There are three reasons for this increase. Firstly, the improvement of surface acidity and hydrophilicity of the material enhances the adsorption of water. The thin layer of adsorbed water promotes the adsorption of NH_3_ through a dissolution effect. Secondly, under humid conditions, water facilitates ionization reactions, forming reactive sites. For example, carboxyl groups can form Brønsted acidic sites, while ammonia can form NH_4_^+^ sites [49]. These acid–base sites can form Brønsted acid–base pairs, thereby enhancing the adsorption capacity of NH_3_ [50]. As shown in Appendix A, the presence of water vapor facilitates the ionization process, promoting substitution reactions between ammonia and organic acids and esters [13], which is beneficial for the formation of organic amines [51]. Among the samples tested, as shown in the Figure 9 ACF-N-6 exhibited a higher ability to increase ammonia adsorption capacity through Brønsted acid–base reactions after the introduction of water vapor. This can be attributed to the higher content of carboxy and lactone groups in ACF-N-6 under dry conditions. The Appendix A provides a schematic illustration of removal of NH_3_ by ACF under dry and wet conditions.

### 3.5. Correlation between Surface Functional Groups and Capacity

To analyze which functional groups play a key role in the adsorption capacity, the adsorption capacities of activated carbon fibers (ACFs) with different functional group contents were measured, and the linear correlation coefficient (R2) was calculated to analyze the correlation between the adsorption capacity and functional group content of the adsorbents. Under dry conditions, the adsorption of ammonia on ACF was found to be related to the total acidic oxygen functional groups (as shown in Figure 10). The results showed that the correlation between hydroxyl and lactone functional groups and the adsorption capacity of ammonia was weaker compared to carboxyl groups. This may be because weak acidic oxygen groups (such as hydroxyl and lactone) combine with ammonia through Lewis acid–base reactions, while strong acidic oxygen groups (carboxyl) react with ammonia to form Brønsted acid–base reactions, indicating that carboxylic acid functional groups significantly affect the adsorption capacity of ammonia [49]. In contrast, phenolic and lactone functional groups have a smaller impact on the adsorption capacity.

## 4. Conclusions

In this study, gelatin-based activated carbon fiber (ACF) was modified by nitric acid impregnation to enhance its adsorption performance for ammonia gas. The results showed that nitric acid modification increased the specific surface area and pore volume of ACF, thereby improving its adsorption capacity. However, at high nitric acid concentrations, the acid can cause damage to the pore walls of ACF, leading to a decrease in specific surface area and pore volume.

Surface chemical characterization revealed that nitric acid modification increased the content of oxygen-containing functional groups on the surface of ACF. Carboxyl groups, lactone groups, and phenolic hydroxyl groups played important roles in the adsorption of ammonia, as they provided active sites for Lewis and Brønsted acid–base reactions with ammonia. Correlation analysis indicated that carboxyl groups were particularly important in the adsorption of ammonia. Furthermore, an increase in RH of the gas stream contributed to the adsorption of ammonia by ACF. Under humid conditions, ammonia can dissolve in water vapor, which promotes reactions between ammonia and the functional groups, thereby enhancing the adsorption capacity. Therefore, under humid conditions, ACF exhibited superior adsorption capacity compared to dry conditions.

The ammonia adsorption isotherms were measured on ACF-raw and HNO_3_-treated ACF at temperatures of 293.15 K, 303.15 K, and 313.15 K, and the thermodynamic parameters were calculated. Due to the exothermic nature of the adsorption process, a higher adsorption capacity was observed at 293.15 K, with the five modified samples’ adsorption capacity at 273 mL·g^−1^, 305 mL·g^−1^, 301 mL·g^−1^, 311 mL·g^−1^, and 301 mL·g^−1^, respectively. It can be observed that the modified activated carbon has a higher adsorption capacity for ammonia. The Langmuir–Freundlich isotherm model provided a good fit to the equilibrium measurements of ammonia adsorption. At 293.15 K and 313.15 K, the heat of adsorption for ammonia was determined using the Langmuir–Freundlich parameters, revealing the coexistence of chemical and physical adsorption between ammonia molecules and ACF.

Adsorption experiments demonstrated a significant increase in the ammonia adsorption capacity of the nitric-acid-modified samples, with ACF-N-6 exhibiting the optimal adsorption capacity (14.08 mmol·L^−1^, 293.15 K). The adsorption performance is significantly better than that of conventional carbon materials, and, compared with other high-performance materials, it has the advantages of simple preparation process and low cost, which has a very great application prospect.

## Figures and Tables

**Figure 1 nanomaterials-13-02857-f001:**
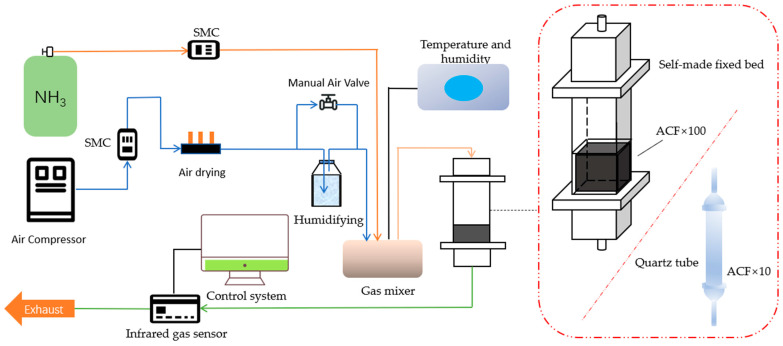
Schematic diagram of NH3 dynamic adsorption experiment.

**Figure 2 nanomaterials-13-02857-f002:**
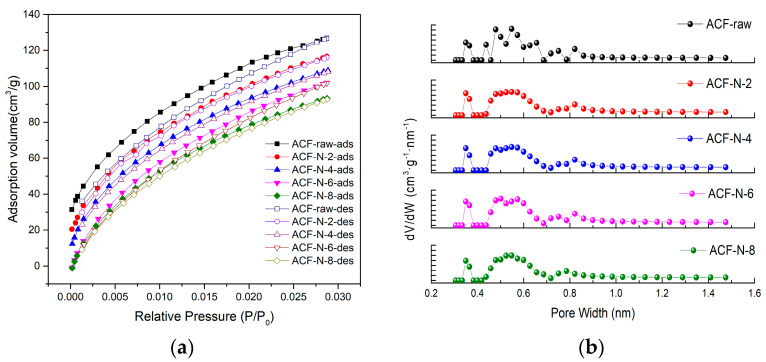
CO_2_ adsorption−desorption isotherms (**a**) and pore−size distribution curves (**b**) of ACF−raw and HNO_3_−treated ACFs.

**Figure 3 nanomaterials-13-02857-f003:**
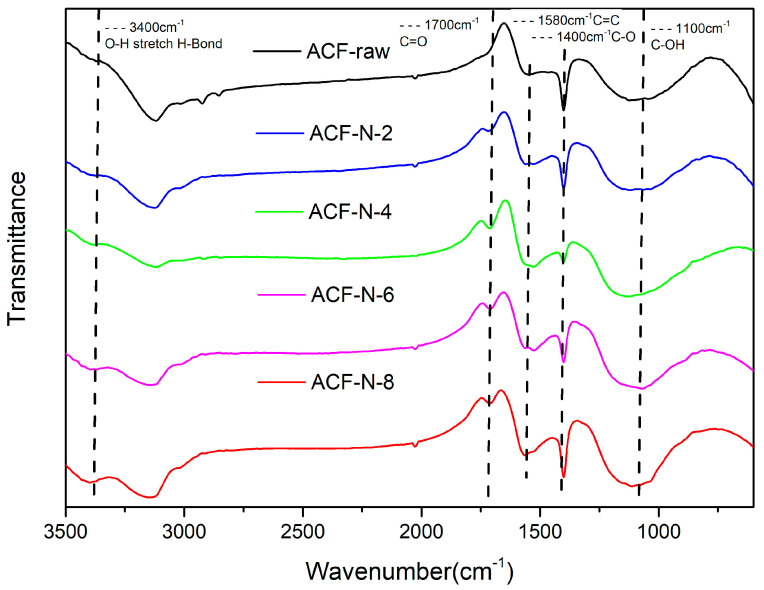
FTIR of ACF series samples.

**Figure 4 nanomaterials-13-02857-f004:**
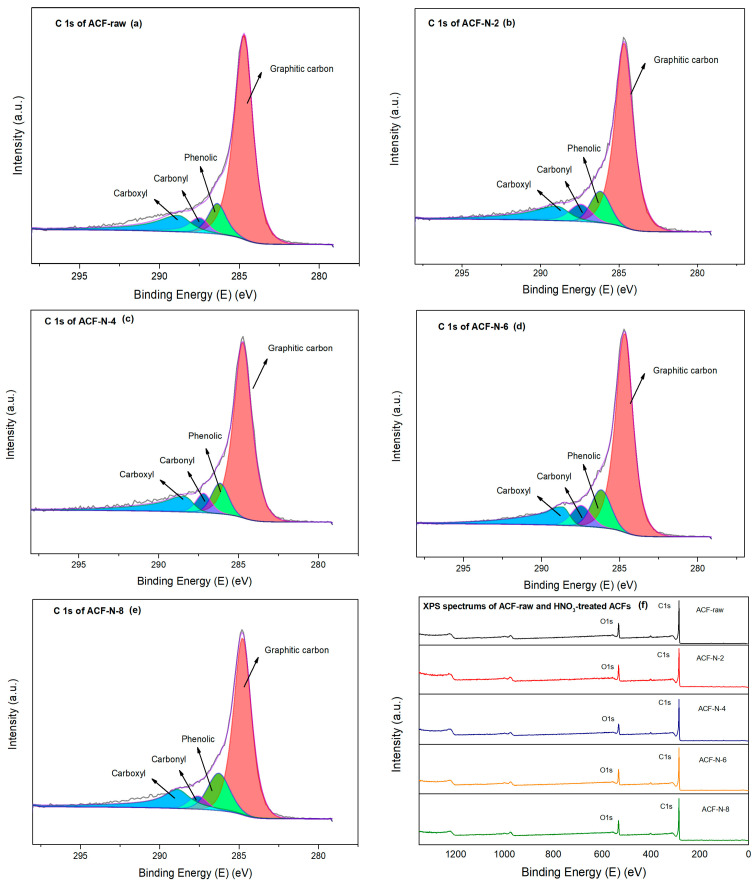
XPS survey spectra of ACF series samples.

**Figure 5 nanomaterials-13-02857-f005:**
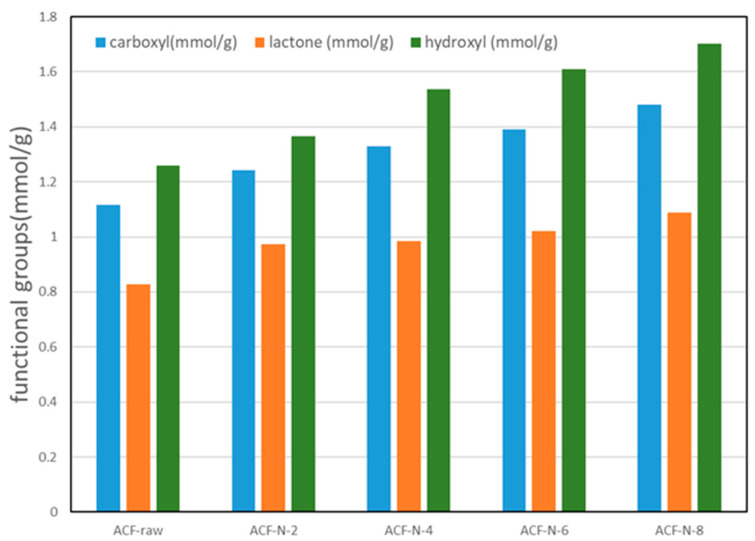
Characterization of surface acidic functional groups through Boehm titration.

**Figure 6 nanomaterials-13-02857-f006:**
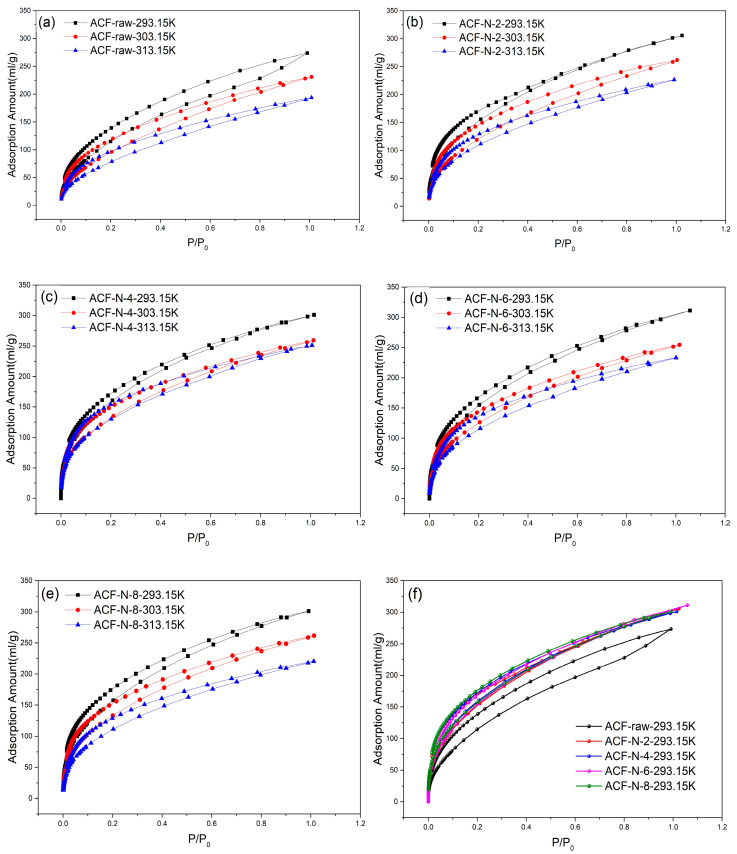
Ammonia adsorption and desorption isotherm ((**a**) ACF-raw; (**b**) ACF-N-2; (**c**) ACF-N-4; (**d**) ACF-N-6; (**e**) ACF-N-8; (**f**) all ACFs at 293.15 K).

**Figure 7 nanomaterials-13-02857-f007:**
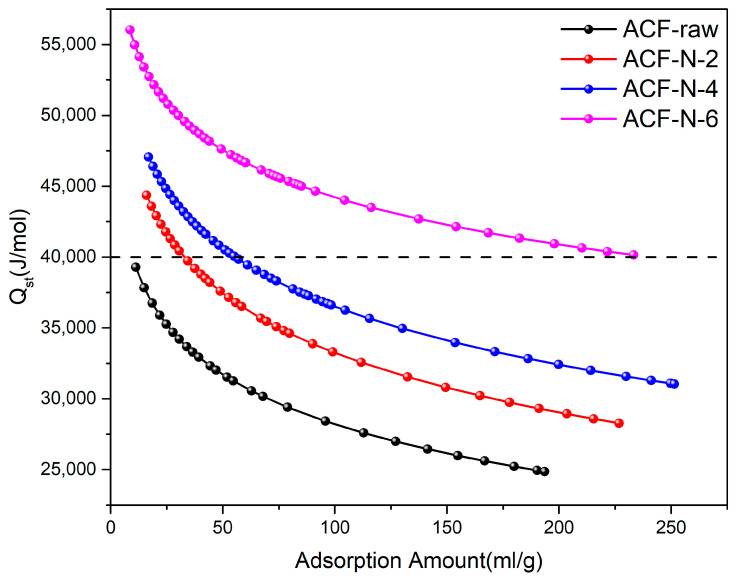
Isosteric heat of ammonia adsorption on ACF-raw and HNO_3_-treated ACF.

**Figure 8 nanomaterials-13-02857-f008:**
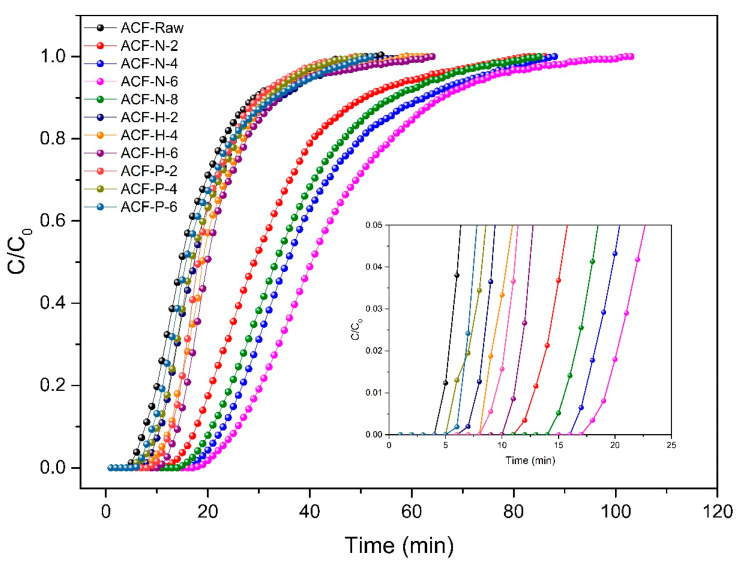
Breakthrough curves of ammonia adsorption by ACF.

**Figure 9 nanomaterials-13-02857-f009:**
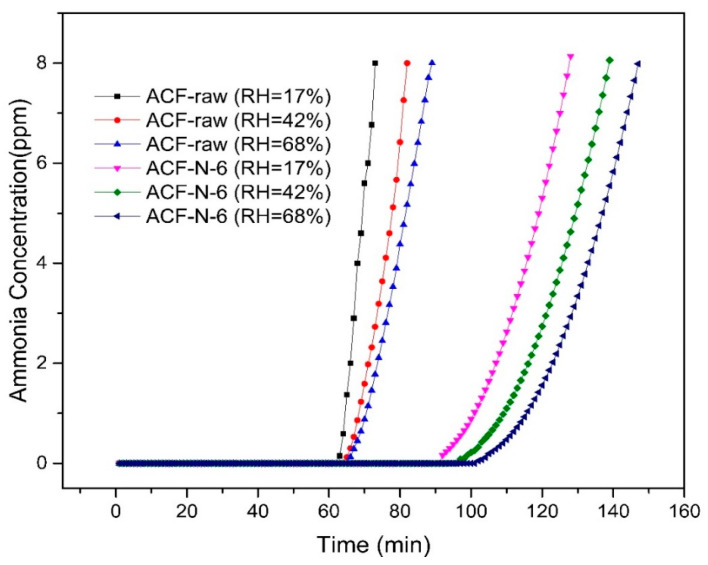
Ammonia adsorption and breakthrough curve in gas streams with different moisture contents.

**Figure 10 nanomaterials-13-02857-f010:**
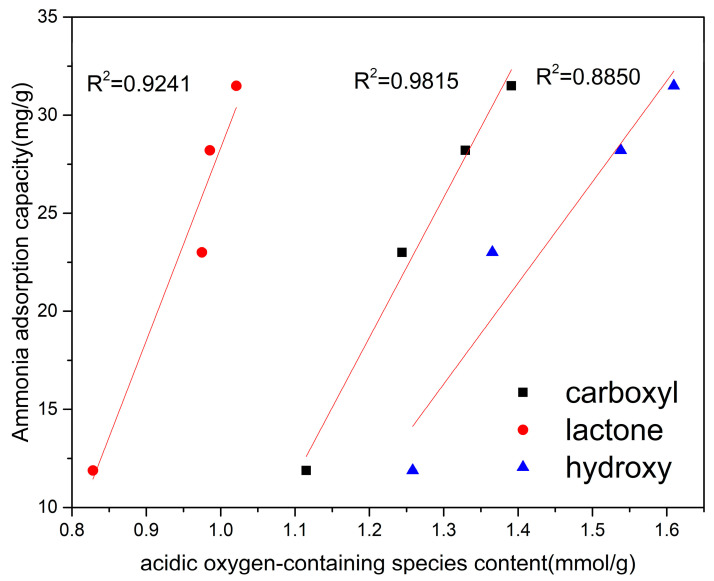
The correlation between the ammonia adsorption capacity and acidic oxygen-containing species content.

**Table 1 nanomaterials-13-02857-t001:** Analysis results based on XPS.

	Deconvolution %				O1s/C1s (%)
	C=C/C-C Peak (284.8 eV)	C-O Peak Phenolics or Ethers (286.2 eV)	C=O Peak Carbonyl (287.5 eV)	O-C=O Peak Carboxylic Acid and Its Derivatives (288.8 eV)	
ACF-raw	71.11	9.90	6.50	12.49	14.72
ACF-N-2	68.16	11.29	8.03	12.51	16.25
ACF-N-4	66.91	13.28	7.21	12.60	18.47
ACF-N-6	65.65	13.48	7.26	13.62	18.92
ACF-N-8	63.36	15.77	4.74	16.13	19.19

**Table 2 nanomaterials-13-02857-t002:** Surface area and pore volume results of ACF-raw and HNO_3_-treated ACF.

Sample	Pore Volume (cm^3^·g^−1^)	Surface Area (m^2^·g^−1^)	d_Pore_ (nm)
ACF-raw	0.304	992.6	0.548
ACF-N-2	0.322	1033.9	0.548
ACF-N-4	0.332	1056.4	0.548
ACF-N-6	0.352	1121.1	0.548
ACF-N-8	0.325	1033.2	0.548

d_pore_: The pore width, accounting for the highest percentage of pores.

**Table 3 nanomaterials-13-02857-t003:** NH_3_ capture performances on various porous materials at 298 K.

Material Type	Sample	NH_3_ Adsorption Capacity (mmol·L^−1^)	Reference
Activated carbon	ACF-N-6	14.08	This study
MOF	Ga-PMOF	10.50	[11]
MOF	Co(NA)2	15.44	[40]
Activated carbon	Doped activated carbon	13.80	[39]
Activated carbon	Mesoporous carbon beads	6.54	[41]
Zeolite	Zeolite-A	8.39	[42]
Biochar	Bamboo biochar	6.74	[43]
Nanostructured materials	[Fe_2_(H_2_opba)_2_Cl_2_(dmso)_2_]·2CH_2_Cl_2_·2dmso	11.28	[44]

**Table 4 nanomaterials-13-02857-t004:** Thermodynamic parameters for NH3 adsorption of ACF.

Sample	Temperature (K)	ΔG (kJ·mol^−1^)	ΔH (kJ·mol^−1^)	ΔS (J·mol^−1^·K^−1^)
ACF-raw	293.15 K	−3.28	−33.36	−102.545
	303.15 K	−2.32		
	313.15 K	−1.23		
ACF-N-6	293.15 K	−6.97	−39.87	−112.372
	303.15 K	−5.72		
	313.15 K	−4.73		

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
