# Peer review of "Efficient Adsorption of Ammonia by Surface-Modified Activated Carbon Fiber Mesh"

_nanomaterials, 2023, doi:10.3390/nano13212857_

Round 1

Reviewer 1 Report

Comments and Suggestions for Authors

Dear Editor and Authors:

Efficient Adsorption of Ammonia by Surface-modified Activated Carbon Fiber Mesh

Overall, although this work presents no big novelty, it presents interesting results. It is indeed, an interesting work and it deserves to be considered for publication in Nanomaterials. However, compared to previous work, it is poorly presented and it lacks some important discussion.  Before I recommend its acceptance, some issues must be clarified and a major revision is needed.

Some other issues that need to be addressed are:

(1)          The novelty of the work should be boldly explicated. The main novelty in this work must be clearly pointed out in the introduction.

(2)          The authors should mention the concept of this work with the progress against the most recent state-of-the-art similar studies.

(3)          The ACF-N-2, ACF-N-4. ACF-N-6 should be properly described and explained.

(4)          Why N2 isotherms were not performed, instead CO2 was? Some research recommend the meaningfulness of CO2 adsorption isotherms for the surface area determination in the ultra-microporosity using BET theory, but perhaps this should be clearly stated in the paper.

(5)          In addition, the BET values should be better discussed with some literature data. Refs. below will help the authors with that.

https://doi.org/10.3390/nano12193480

https://doi.org/10.1016/j.jece.2021.105865

(6)          In Table 2. Analysis results based on XPS, C1s does not mean “graphite” there are many C states. Update the information in Table 2.

(7)          Figures 10 to 13 in a row look extremely messy. The paper should be better organized and presented.

(8)          13 figures and 4 tables look too much. Some figures should be placed as supplementary material.

(9)          Page 18, line 425: “Under dry conditions, the adsorption of ammonia on ACF was found to be related to the total acidic oxygen functional groups (as shown in Figure 13).”….Figure 13 is being mentioned before figure 12? Perhaps these 2 figures could be merged in one.

(10)      Page 16: “……with ammonia to form Brønsted acid-base reactions (Figure 12), indicating that carboxylic acid functional groups significantly affect the adsorption capacity of ammonia…this statement need to be referenced.

(11)      What about the regeneration and reusability of the materials? Can it be done, not in all but in the one with the best performance.

Dear Editor, I believe that if the authors consider and perform all my suggestions/recommendations this manuscript would improve a lot and would fit into the high-standard papers published by Nanomaterials.

Author Response

(1)          The novelty of the work should be boldly explicated. The main novelty in this work must be clearly pointed out in the introduction.

We have made changes in the manuscript.

Based on the current lack of work on the modification of activated carbon fiber mesh for electrothermal regeneration, this study improved the adsorption capacity of viscose-based ACFM for ammonia and further optimized the capacity of the ACF adsorption-electrothermal regeneration unit for ammonia, which contributes to the achievement of efficient and durable protection against ammonia.

(2)          The authors should mention the concept of this work with the progress against the most recent state-of-the-art similar studies.

We have added studies related to the modification of activated carbon fibres to increase the content of acidic oxygenated functional groups on the surface as well as studies related to electrothermal regeneration.

Chen[18] et al. oxidatively modified ACFC by nitric acid to increase the surface acidic oxygenated functional group content and electrically regenerated. Qajar et al. [25] reported that the NH3 adsorption capacity of synthetic non-porous carbon material increased from 180 mg·g-1 to 306 mg·g-1 after treatment with nitric acid at 298K. In 2016, Zheng et al. [26] prepared ACF composite materials from phenolic precursors, and the NH3 adsorption capacity increased from 10 mg·g-1 to 50 mg·g-1 after oxidation with concentrated HNO3. Xu et al. [15] prepared a series of ACF membrane adsorbents simultaneously loaded with metal chlorides and metal oxides, achieving an NH3 adsorption capacity of 22.5 mg·g-1.

(3)          The ACF-N-2, ACF-N-4. ACF-N-6 should be properly described and explained.

We have made changes in the manuscript.

The unmodified ACF is labelled ACF-raw, and the material obtained by acid impregnation is labelled ACF-X-A, where X shows the acid used for impregnation (N stands for nitric acid, H stands for hydrochloric acid, and P stands for phosphoric acid), and A shows the concentration of nitric acid used for impregnation (mol-L-1), such as the sample obtained by using 6 mol-L-1 nitric acid in the modification is labelled ACF-N-6.

(4)          Why N2 isotherms were not performed, instead CO2 was? Some research recommend the meaningfulness of CO2 adsorption isotherms for the surface area determination in the ultra-microporosity using BET theory, but perhaps this should be clearly stated in the paper.

We have made changes in the manuscript.

The surface area and pore characteristics of materials are often determined by N2 adsorption/desorption isotherms at 77 K using the Brunauer-Emmett-Teller (BET) theory and density functional theory methods (DFT). However, there are kinetic limitations at low temperatures (77 K), where N2 molecules have difficulty diffusing into ultra-micropores and smaller micropores within a short period of time, leading to deviations in the analysis results. On the other hand, CO2 molecules have a smaller kinetic diameter and a very high saturation vapor pressure at 273 K. In this condition, gas diffusion is faster, and CO2 molecules can enter micropores below 0.4 nm. Therefore, CO2 has become an effective probing molecule for studying extremely narrow micropores in carbon materials.

(5)          In addition, the BET values should be better discussed with some literature data.

We have made changes in the manuscript.

The CO2 adsorption-desorption isotherms(a) and pore size distribution(b) of ACF-raw and HNO3-treated ACFs are shown in Figure 2. At 273K, the curve depicting the changes in the amount of adsorbed CO2 with respect to pressure exhibits the characteristics of an I-type isotherm, as per the classification established by IUPAC. An I-type isotherm is typically indicative of an adsorbent material with a microporous structure. Figure 2b illustrates that the pore structure of all samples predominantly consisted of micropores. The primary pore size distribution of ACF ranged from 0.3 to 1.0 nm, with a pore size of 0.548 nm accounting for the highest percentage of pores. The specific surface area of micropores in ACF varied from 992.6 to 1056.4 m2·g-1 After HNO3 impregnation, the specific surface area and micropore volume of ACF slightly increase. With the increase in nitric acid concentration, the micropore volume gradually increases and there is no significant change in the average pore size. However, when the concentration reaches 8 mol·L-1, the micropore volume starts to decrease. This could be due to the strong oxidizing nature of HNO3, which may cause damage to the pore walls of ACF, thereby reducing the specific surface area and pore volume, and affecting the adsorption performance. Therefore, it is important to choose an appropriate concentration of the oxidizing agent during the modification process.

(6)          In Table 2. Analysis results based on XPS, C1s does not mean “graphite” there are many C states. Update the information in Table 2.

We have made changes in the manuscript.

Deconvolution %

O1s/C1s(%)

C=C/C-C peak

(284.8eV)

C-O peak

phenolics or ethers (286.2eV)

C=O peak

carbonyl (287.5eV)

O-C=O peak

Carboxylic acid and its derivatives

(288.8eV)

ACF-raw

71.11

9.90

6.50

12.49

14.72

ACF-N-2

68.16

11.29

8.03

12.51

16.25

ACF-N-4

66.91

13.28

7.21

12.60

18.47

ACF-N-6

65.65

13.48

7.26

13.62

18.92

ACF-N-8

63.36

15.77

4.74

16.13

19.19

(7)          Figures 10 to 13 in a row look extremely messy. The paper should be better organized and presented.

Figures 10 and 12 are deleted and can be found in the support information.

(8)          13 figures and 4 tables look too much. Some figures should be placed as supplementary material.

Figures 10 and 12 are deleted and can be found in the support information.

(9)          Page 18, line 425: “Under dry conditions, the adsorption of ammonia on ACF was found to be related to the total acidic oxygen functional groups (as shown in Figure 13).”….Figure 13 is being mentioned before figure 12? Perhaps these 2 figures could be merged in one.

We have made changes in the manuscript.

To analyze which functional groups play a key role in the adsorption capacity, the adsorption capacities of activated carbon fibers (ACF) with different functional group contents were measured, and the linear correlation coefficient (R2) was calculated to analyze the correlation between the adsorption capacity and functional group content of the adsorbents. Under dry conditions, the adsorption of ammonia on ACF was found to be related to the total acidic oxygen functional groups (as shown in Figure 10). The results showed that the correlation between hydroxyl and lactone functional groups and the adsorption capacity of ammonia was weaker compared to carboxyl groups.

(10)      Page 16: “……with ammonia to form Brønsted acid-base reactions (Figure 12), indicating that carboxylic acid functional groups significantly affect the adsorption capacity of ammonia…this statement need to be referenced.

We have made changes in the manuscript.

This may be because weak acidic oxygen groups (such as hydroxyl and lactone) combine with ammonia through Lewis acid-base reactions, while strong acidic oxygen groups (carboxyl) react with ammonia to form Brønsted acid-base reactions, indicating that carboxylic acid functional groups significantly affect the adsorption capacity of ammonia[49]. In contrast, phenolic and lactone functional groups have a smaller impact on the adsorption capacity.

(11)      What about the regeneration and reusability of the materials? Can it be done, not in all but in the one with the best performance.

For adsorption saturated ACF-N-6, it was baked at 110°C for 6h, cooled and then subjected to ammonia adsorption in a homemade activated carbon fiber adsorption bed. The data from the adsorption and desorption cycles are shown in Figure. The adsorption capacity gradually decreased with increasing number of cycles and stabilized after a 40% decrease. After increasing the desorption temperature to 150°C, the adsorption capacity decreased by 27% compared to the initial state. Research on electrothermal regeneration of this material is still underway, with pre-experimentation through conventional thermal desorption, and the thermal energy provided by electrothermal regeneration will be utilized more completely, so that a large degree of regeneration of this material can be achieved.

Reviewer 2 Report

Comments and Suggestions for Authors

The manuscript titled « Efficient Adsorption of Ammonia by Surface-modified Activated Carbon Fiber Mesh “aimed on the synthesis and characterization of viscose-based activated carbon fiber for adsorption of ammonia. This work presents some interesting results on the adsorption of ammonia pollutants, and minor revision is requested before acceptation:

1)      In the introduction part, authors could add and discuss some other published works on the ACF utilization for the removal of NH3 from liquid medias. Please discuss the chemical that used for the adsorption of NH3 over ACF.

2)      Figure.3: Is it a Transmittance or Absorbance? please check it.

3)      Figure.7 should be improved and the indexing should be removed and noted in the title.

4)      Comparison on the adsorption capacity of NH3 over the prepared materials and other materials published elsewhere should be added to show the originality of this work.

5)      It will be better if authors reduce the figure number from 13 to 10

Author Response

Comments and Suggestions for Authors

The manuscript titled « Efficient Adsorption of Ammonia by Surface-modified Activated Carbon Fiber Mesh “aimed on the synthesis and characterization of viscose-based activated carbon fiber for adsorption of ammonia. This work presents some interesting results on the adsorption of ammonia pollutants, and minor revision is requested before acceptation:

First of all, thank you very much for your professional comments on this article. We checked the overall manuscript carefully and gave our response. The changes have been marked in red within the revised manuscript. We look forward to your professional comments. Thanks again for your guidance.

1)      In the introduction part, authors could add and discuss some other published works on the ACF utilization for the removal of NH3 from liquid medias. Please discuss the chemical that used for the adsorption of NH3 over ACF.

There has been limited research on the use of modified activated carbon fibers for the removal of ammonia nitrogen in water. Most studies in this area primarily focus on utilizing modified zeolites for the treatment of ammonia nitrogen in water.

We have discussed the chemical that used for the adsorption of NH3 over ACF.

2)      Figure.3: Is it a Transmittance or Absorbance? please check it.

It is a Transmittance. The original graph was absorbance, which has been changed to transmittance, which has been corrected in the graph.

3)      Figure.7 should be improved and the indexing should be removed and noted in the title.

We have put in Table 7 with support information and made changes.

4)      Comparison on the adsorption capacity of NH3 over the prepared materials and other materials published elsewhere should be added to show the originality of this work.

We increased the adsorption properties of different materials for ammonia and compared them.

Material Type

Sample

NH3 Adsorption Capacity

(mmol·L-1)

Reference

Activated carbon

ACF-N-6

14.08

This study

MOF

Ga-PMOF

10.50

[11]

MOF

Co(NA)2

15,44

[41]

Activated carbon

doped activated carbon

13.80

[40]

Activated carbon

mesoporous carbon beads

6.54

[42]

Zeolite

Zeolite-A

8.39

[43]

Biochar

bamboo biochar

6.74

[44]

nanostructured materials

[Fe2(H2opba)2Cl2(dmso)2]·2CH2Cl2·2dmso

11.28

[45]

5)      It will be better if authors reduce the figure number from 13 to 10

We have reduced the figure number from 13 to 10.

Reviewer 3 Report

Comments and Suggestions for Authors 1. Table 1. Pore volume unit must be corrected - cm3*g-1. Please explain how the pore width value of 0.548 nm was chosen as the PSD curves show that all samples have a wide pore size distribution. 2. Line 113. According to the IUPAC recommendations, porous structure parameters, such as BET surface, of microporous adsorbents are usually calculated from N2 adsorption isotherms at 77 K and CO2/273K adsorption is used for estimation of Ultramicroporous parameters (pores narrower 1,0 nm). Moreover, saturation pressure of CO2 at 273 K is over 3,4 MPa therefore the point of P/P0=1 is usually unachievable while using automatic adsorption devices such as Nova4000. However, authors show full isotherms of CO2 adsorption up to saturation pressure (P/P0=1) on the Figure 2. Does it mean that saturation pressure of CO2 has been really reached within the experiment? How? Please, give detailed information on the parameters of the adsorption experiment in the section 2.3. 3. Figure 2. The appearance of the adsorption-desorption and PSD curves is very poor. It is recommended to shift the curves vertically and (for 2a) designate adsorption and desorption branches with different symbols. 4. Figure 6. Please explain in the text the origin and the nature of the hysteresis loops occurring within adsorption and desorption processes of ammonia on the samples. 5. It is recommended to improve the appearance of the picture. Maybe split it into two parts. What is “A” in the caption of the OX axis (P/P0)? Comments on the Quality of English Language

Minor editing of English language required

Author Response

First of all, thank you very much for your professional comments on this article. We checked the overall manuscript carefully and gave our response. The changes have been marked in red within the revised manuscript. We look forward to your professional comments. Thanks again for your guidance.

  1. Table 1. Pore volume unit must be corrected - cm3*g-1. Please explain how the pore width value of 0.548 nm was chosen as the PSD curves show that all samples have a wide pore size distribution.

The pore width value of 0.548 nm is the pore size with the largest pore volume i.e. the largest percentage of micropores, and changes have been made in the manuscript.

  1. Line 113. According to the IUPAC recommendations, porous structure parameters, such as BET surface, of microporous adsorbents are usually calculated from N2 adsorption isotherms at 77 K and CO2/273K adsorption is used for estimation of Ultra microporous parameters (pores narrower 1,0 nm). Moreover, saturation pressure of CO2 at 273 K is over 3,4 MPa therefore the point of P/P0=1 is usually unachievable while using automatic adsorption devices such as Nova4000. However, authors show full isotherms of CO2 adsorption up to saturation pressure (P/P0=1) on the Figure 2. Does it mean that saturation pressure of CO2 has been really reached within the experiment? How? Please, give detailed information on the parameters of the adsorption experiment in the section 2.3.

the point of P/P0=1 is unachievable while using automatic adsorption devices such as Nova4000. Data-processing errors, which have been corrected in the manuscript.

  1. Figure 2. The appearance of the adsorption-desorption and PSD curves is very poor. It is recommended to shift the curves vertically and (for 2a) designate adsorption and desorption branches with different symbols.

As shown in the picture above, we have made changes to the image.

  1. Figure 6. Please explain in the text the origin and the nature of the hysteresis loops occurring within adsorption and desorption processes of ammonia on the samples.

As a result of chemical adsorption, some of the ammonia adsorbed cannot be desorbed by altering the pressure at this temperature. This leads to a lower amount of desorbed ammonia than the amount adsorbed, resulting in the occurrence of hysteresis loops. The presence of hysteresis loops indicates that the adsorption of ammonia by ACF occurs through both physical and chemical adsorption mechanisms.

  1. It is recommended to improve the appearance of the picture. Maybe split it into two parts. What is “A” in the caption of the OX axis (P/P0)?

This mistake was made while making the drawing, and we have corrected it in the manuscript.

Round 2

Reviewer 1 Report

Comments and Suggestions for Authors

The authors have addressed all my comments properly. I know able to accept it.

Reviewer 2 Report

Comments and Suggestions for Authors

Now the revised manuscript can be accepted as is

Reviewer 3 Report

Comments and Suggestions for Authors

Accept.